# POINTNET WITH KAN VERSUS POINTNET WITH MLP FOR 3D CLASSIFICATION AND SEGMENTATION OF POINT SETS

## ABSTRACT

Kolmogorov–Arnold Networks (KANs) have recently gained attention as an alternative to traditional Multilayer Perceptrons (MLPs) in deep learning frameworks. KANs have been integrated into various deep learning architectures such as convolutional neural networks, graph neural networks, and transformers, with their performance evaluated. However, their effectiveness within point-cloud-based neural networks remains unexplored. To address this gap, we incorporate KANs into PointNet for the first time to evaluate their performance on 3D point cloud classification and segmentation tasks. Specifically, we introduce PointNet-KAN, built upon two key components. First, it employs KANs instead of traditional MLPs. Second, it retains the core principle of PointNet by using shared KAN layers and applying symmetric functions for global feature extraction, ensuring permutation invariance with respect to the input features. In traditional MLPs, the goal is to train the weights and biases with fixed activation functions; however, in KANs, the goal is to train the activation functions themselves. We use Jacobi polynomials to construct the KAN layers. We extensively and systematically evaluate PointNet-KAN across various polynomial degrees and special types such as the Lagrange, Chebyshev, and Gegenbauer polynomials. Our results show that PointNet-KAN achieves competitive performance compared to PointNet with MLPs on benchmark datasets for 3D object classification and segmentation, despite employing a shallower and simpler network architecture. We hope this work serves as a foundation and provides guidance for integrating KANs, as an alternative to MLPs, into more advanced point cloud processing architectures.

## 1 INTRODUCTION

Kolmogorov-Arnold Networks (KANs), introduced by Liu et al. (2024), have recently emerged as an alternative modeling framework to traditional Multilayer Perceptrons (MLPs) (Cybenko, 1989; Hornik et al., 1989). KANs are based on the Kolmogorov-Arnold representation theorem (Kolmogorov, 1957; Arnold, 2009). Unlike MLPs, which rely on fixed activation functions while training weights and biases, the objective in KANs is to train the activation functions themselves (Liu et al., 2024).

The performance of KANs has been evaluated across various domains, including scientific machine learning tasks (Wang et al., 2024b; Shukla et al., 2024; Abueidda et al., 2024; Koenig et al., 2024), image classification (Azam & Akhtar, 2024; Cheon, 2024; Lobanov et al., 2024; Yu et al., 2024; Tran et al., 2024), image segmentation (Li et al., 2024; Tang et al., 2024), image detection (Wang et al., 2024a), audio classification (Yu et al., 2024), and other applications. Additionally, from a neural network architecture perspective, KANs have been integrated into convolutional neural networks (CNNs) (Azam & Akhtar, 2024; Bodner et al., 2024) and graph neural networks (Kiamari et al., 2024; Bresson et al., 2024; Zhang & Zhang, 2024; De Carlo et al., 2024).

However, the efficiency of KANs for 3D point cloud data has not yet been explored. Point cloud data plays a critical role in various domains, including computer graphics, computer vision, robotics, and autonomous driving (Uy & Lee, 2018; Li et al., 2020; Guo et al., 2020; Zhang et al., 2023a;b). One of the most successful neural networks for deep learning on point cloud data is PointNet, introduced

by Qi et al. (2017a). Following this, several modified and advanced versions of PointNet have been developed (Qi et al., 2017b; Shen et al., 2018; Thomas et al., 2019; Wang et al., 2019; Zhao et al., 2021). To the best of our knowledge, the only existing work embedding KANs into PointNet involves 2D supervised learning in the context of computational fluid dynamics (Kashefi, 2024). In this work, we integrate KANs into PointNet for the first time to evaluate its performance on classification and segmentation tasks for 3D point cloud data.

It is important to clarify that by embedding KANs into PointNet, we do not simply mean replacing every instance of MLPs with KANs. While such an approach could be considered a research case, our goal is to preserve and utilize the core principles upon which PointNet is built. First, we apply shared KANs, meaning that the same KANs are applied to all input points. Second, we utilize a symmetric function, such as the max function, to extract global features from the points. These two elements are fundamental to PointNet, and by maintaining them, we ensure that the network remains invariant to input permutations. Our objective is to propose a version of PointNet integrated with KANs that retains these two essential properties, which we refer to as PointNet-KAN throughout the rest of this article. Moreover, we focus on PointNet (Qi et al., 2017a) rather than more advanced versions (Qi et al., 2017b; Shen et al., 2018; Thomas et al., 2019; Wang et al., 2019; Zhao et al., 2021) to directly and explicitly investigate the effect of KANs on the network's performance. Using more complex versions of PointNet could introduce other factors that might obscure the direct influence of KANs, making it challenging to determine whether any performance changes are due to the KAN architecture or other components of the network.

We use Jacobi polynomials to construct PointNet-KAN and investigate its performance across different polynomial degrees. Additionally, we examine the effect of special cases of Jacobi polynomials, including Legendre polynomials, Chebyshev polynomials of the first and second kinds, and Gegenbauer polynomials. The performance of PointNet-KAN is evaluated across classification and part segmentation tasks. Overall, the summary of our key contributions is as follows:

- We introduce PointNet with KANs (i.e., PointNet-KAN) for the first time and evaluate its performance against PointNet with MLPs.

- We embed KAN into a point-cloud-based neural network for the first time, for computer vision tasks on unordered 3D point sets.

- We conduct an extensive evaluation of the hyperparameters of PointNet-KAN, specifically the degree and type of polynomial used in constructing KANs.

- We assess the efficiency of PointNet-KAN on benchmarks for 3D object classification and segmentation tasks.

- We demonstrate that PointNet-KAN achieves competitive performance to PointNet, despite having a much shallower and simpler network architecture.

- We release our code to support reproducibility and future research.

## 2 Kolmogorov-Arnold Network (KAN) layers

Inspired by the Kolmogorov-Arnold representation theorem (Kolmogorov, 1957; Arnold, 2009), Kolmogorov-Arnold Network (KAN) has been proposed as a novel neural network architecture by Liu et al. (2024). According to the theorem, multivariate continuous function can be expressed as a finite composition of continuous univariate functions and additions. To describe the structure of KAN straightforwardly, consider a single-layer KAN. The network's input is a vector $\boldsymbol{r}$ of size $d_{\text{input}}$, and its output is a vector $\boldsymbol{s}$ of size $d_{\text{output}}$. In this configuration, the single-layer KAN maps the input to the output as follows:

$$\boldsymbol{s}_{d_{\text{output}}} = \mathbf{A}_{d_{\text{output}} \times d_{\text{input}}} \boldsymbol{r}_{d_{\text{input}}}, \tag{1}$$

where the tensor $\mathbf{A}_{d_{\text{output}} \times d_{\text{input}}}$ is expressed as:

$$\mathbf{A}_{d_{\text{output}} \times d_{\text{input}}} = \begin{bmatrix} \psi_{1,1}(\cdot) & \psi_{1,2}(\cdot) & \cdots & \psi_{1,d_{\text{input}}}(\cdot) \\ \psi_{2,1}(\cdot) & \psi_{2,2}(\cdot) & \cdots & \psi_{2,d_{\text{input}}}(\cdot) \\ \vdots & \vdots & \ddots & \vdots \\ \psi_{d_{\text{output}},1}(\cdot) & \psi_{d_{\text{output}},2}(\cdot) & \cdots & \psi_{d_{\text{output}},d_{\text{input}}}(\cdot) \end{bmatrix}, \tag{2}$$

where each $\psi(\gamma)$ (the subscript is removed to lighten notation) is defined as:

$$\psi(\gamma) = \sum_{i=0}^{n} \omega_i f_i^{(\alpha,\beta)}(\gamma), \tag{3}$$

where $f_i^{(\alpha,\beta)}(\gamma)$ represents the Jacobi polynomial of degree $i$, $n$ is the polynomial order of $\psi$, and $\omega_i$ are trainable parameters. Hence, the total number of trainable parameters embedded in $\mathbf{A}$ is $(n+1) \times d_{\text{input}} \times d_{\text{output}}$. We implement $f_n^{(\alpha,\beta)}(\gamma)$ using a recursive relation (Szegő, 1939):

$$f_n^{(\alpha,\beta)}(\gamma) = (a_n\gamma + b_n)f_{n-1}^{(\alpha,\beta)}(\gamma) + c_n f_{n-2}^{(\alpha,\beta)}(\gamma), \tag{4}$$

where the coefficients $a_n$, $b_n$, and $c_n$ are given by:

$$a_n = \frac{(2n+\alpha+\beta-1)(2n+\alpha+\beta)}{2n(n+\alpha+\beta)}, \tag{5}$$

$$b_n = \frac{(2n+\alpha+\beta-1)(\alpha^2-\beta^2)}{2n(n+\alpha+\beta)(2n+\alpha+\beta-2)}, \tag{6}$$

$$c_n = \frac{-2(n+\alpha-1)(n+\beta-1)(2n+\alpha+\beta)}{2n(n+\alpha+\beta)(2n+\alpha+\beta-2)}, \tag{7}$$

with the following initial conditions:

$$f_0^{(\alpha,\beta)}(\gamma) = 1, \tag{8}$$

$$f_1^{(\alpha,\beta)}(\gamma) = \frac{1}{2}(\alpha+\beta+2)\gamma + \frac{1}{2}(\alpha-\beta). \tag{9}$$

Since $f_n^{(\alpha,\beta)}(\gamma)$ is recursively constructed, the polynomials $f_i^{(\alpha,\beta)}(\gamma)$ for $0 \le i \le n$ are computed sequentially. Additionally, because the input to the Jacobi polynomials must lie within the interval $[-1,1]$, the input vector $\mathbf{r}$ needs to be scaled to fit this range before being passed to the KAN layer. To achieve this, we apply the hyperbolic tangent function. Finally, setting $\alpha = \beta = 0$ yields the Legendre polynomial (Abramowitz, 1974; Szegő, 1939), while the Chebyshev polynomials of the first and second kinds are obtained with $\alpha = \beta = -0.5$ and $\alpha = \beta = 0.5$, respectively (Abramowitz, 1974; Szegő, 1939). Additionally, the Gegenbauer (or ultraspherical) polynomials arise when $\alpha = \beta$ (Szegő, 1939).

## 3 OVERVIEW OF POINTNET AND ITS KEY PRINCIPLES

Consider a point cloud $\mathcal{X}$ as an unordered set with $N$ points, defined as $\mathcal{X} = \left\{\mathbf{x}_j \in \mathbb{R}^d\right\}_{j=1}^{N}$. The dimension (or number of features) of each $\mathbf{x}_j$ is shown by $d$. According to the Theorem 1 proposed in Qi et al. (2017a), a set function $g : \mathcal{X} \to \mathbb{R}$ can be defined to map this set of points to a vector as follows:

$$g(\mathbf{x}_1, \mathbf{x}_2, \ldots, \mathbf{x}_N) = \tau\left(\max_{j=1,\ldots,N}\{h(\mathbf{x}_j)\}\right), \tag{10}$$

Table 1: Classification results on ModelNet40 (Wu et al., 2015). In PointNet-KAN, the Jacobi polynomial degree is set to 4 (i.e., $n = 4$) with $\alpha = \beta = 1.0$. Time complexity for PointNet-KAN and PointNet is provided. 'M' stands for million.

|  | normal vector | number of points | Mean class accuracy | Overall accuracy | FLOPs/sample |
|---|---|---|---|---|---|
| PointNet++ (Qi et al., 2017b) | no | 2048 | - | 90.7 | - |
| PointNet++ (Qi et al., 2017b) | yes | 2048 | - | 91.9 | - |
| DGCNN (Wang et al., 2019) | no | 2048 | 90.7 | 93.5 | - |
| Point Transformer (Zhao et al., 2021) | yes | - | 90.6 | 93.7 | - |
| PointMLP (Ma et al., 2022) | no | 1000 | 91.4 | 94.5 | - |
| ShapeLLM (Qi et al., 2024) | no | 1000 | 94.8 | 95.0 | - |
| PointNet (baseline) (Qi et al., 2017a) | no | 1024 | 72.6 | 77.4 | 148M |
| PointNet (Qi et al., 2017a) | no | 1024 | 86.2 | 89.2 | 440M |
| PointNet-KAN | no | 1024 | 82.7 | 87.5 | 60M |
| PointNet-KAN | yes | 1024 | 87.2 | 90.5 | 110M |

where $\max$ is a vector-wise max operator that takes $N$ vectors as input and returns a new vector, computed as the element-wise maximum. In PointNet (Qi et al., 2017a), the continuous functions $\tau$ and $h$ are implemented as MLPs. In this work, we replace $\tau$ and $h$ with KANs, resulting in PointNet-KAN. Note that the function $g$ is invariant to the permutation of input points. Details of this theorem and its proof can be found in Qi et al. (2017a).

## 4 ARCHITECTURE OF POINTNET-KAN

**Classification branch** The top panel of Fig. 1 demonstrates the classification branch of PointNet-KAN. The architecture of the classification branch is explained as follows. The PointNet-KAN model accepts input with dimensionality corresponding to 3D spatial coordinates (i.e., $d = 3$) and possibly the 3D normal vector as part of the point set representation (i.e., $d = 6$). A shared KAN layer maps the input feature vector from its original space to an intermediate feature space of dimension 3072. Following the first shared KAN layer, batch normalization (Ioffe & Szegedy, 2015) is applied. After normalization, a max pooling operation is performed to extract global features by computing the maximum value across all points in the point cloud. Next, the global feature is passed through a KAN layer, which reduces the dimensionality to the number of output channels (i.e., $k$), corresponding to the classification task. A softmax function is applied to the output to convert the logits into class probabilities. The concept of shared KANs is analogous to the shared MLPs used in PointNet (Qi et al., 2017a). It means that the same functional tensor, **A**, is applied uniformly to the input or intermediate features in PointNet-KAN. The use of the shared KAN layers and the symmetric max-pooling function ensure that PointNet-KAN is invariant to the order of the points in the point cloud.

**Part segmentation branch** As shown in the bottom panel of Fig. 1, the part segmentation branch of the PointNet-KAN is described as follows. The input is first passed through a shared KAN layer, transforming it to an intermediate feature space of size 640, followed by batch normalization. These local features are then processed by a second shared KAN layer, mapping them to a higher-dimensional space of size 5120, and another batch normalization step is applied. A max pooling operation extracts a global feature representing the entire point cloud, which is then expanded to match the number of points. The one-hot encoded class label, representing the object category, is concatenated with the local features and the global feature. This combined feature, consisting of local features of size 640, global features of size 5120, and the class label, is passed through a shared KAN layer to reduce the feature size to 640, followed by batch normalization. A final shared KAN layer generates the output, delivering point-wise segmentation predictions, followed by a softmax function to convert the logits into class probabilities.

**Classification Network**

**Segmentation Network**

Figure 1: Architecture of PointNet-KAN. The classification network is shown in the top panel, and the segmentation network is shown in the bottom panel. $N$ is the number of points in a point cloud. $d$ indicates the number of input point features (e.g., spatial coordinates, normal vectors, etc.). $k$ indicates the number of classes (e.g., for the ModelNet40 (Wu et al., 2015) benchmark, $k = 40$; see Sect. 5.1). $m$ indicates the total number of possible parts (e.g., for the ShapeNet part (Yi et al., 2016) benchmark, $m = 50$; see Sect. 5.2).

# 5 EXPERIMENT AND DISCUSSION

## 5.1 3D OBJECT CLASSIFICATION

We evaluate PointNet-KAN on the ModelNet40 (Wu et al., 2015) shape classification benchmark, which contains 12,311 models across 40 categories, with 9,843 models allocated for training and 2,468 for testing. Similar to Qi et al. (2017a), we uniformly sample 1,024 points from the mesh faces and normalize them into a unit sphere. We also conduct an experiment with included normal vectors as input features, computed using the trimesh library (Dawson-Haggerty et al.). Table 1 presents the classification results of PointNet-KAN, with a polynomial degree of 4 (i.e., $n = 4$ in Eq. 3) and $\alpha = \beta = 1$. Training details are provided in A.1. The obtained results can be interpreted from two perspectives.

First, comparing PointNet-KAN with PointNet (baseline) (Qi et al., 2017a) and PointNet (Qi et al., 2017a) shows that PointNet-KAN (with or without normal vectors) achieves higher accuracy than PointNet (baseline). Additionally, PointNet-KAN with normal vectors as input features outperforms PointNet. The number of trainable parameters for PointNet-KAN with $n = 4$, PointNet (baseline), and PointNet in the classification branch is approximately 1M, 0.8M, and 3.5M, respectively. It is worth noting that PointNet-KAN with $n = 2$ has only roughly 0.6M trainable parameters, making it lighter than PointNet (baseline) while still achieving an overall accuracy of 89.9 (see Table 4). Notably, despite its simpler architecture—lacking the input and feature transforms found in Point-Net, as shown in Fig. 2 of Qi et al. (2017a), and having only 3 hidden layers compared to the 8 hidden layers of PointNet—PointNet-KAN still delivers competitive results, with overall accuracy of 90.5% versus 89.2%. From a time complexity perspective, the number of floating-point operations required for one forward pass of the PointNet-KAN model is significantly lower than that of PointNet, as shown in Table 1.

From the second perspective, we observe that other advanced point-cloud-based deep learning frameworks, such as PointNet++ (Qi et al., 2017b), DGCNN (Wang et al., 2019), Point Transform (Zhao et al., 2021), PointMLP (Ma et al., 2022), and ShapeLLM (Qi et al., 2024), outperform PointNet-KAN, as listed in Table 1, though these models employ more advanced and complex architectures involving MLPs. This raises the question of whether redesigning these networks using KANs instead of MLPs could improve their accuracy. While the current article focuses on evaluating

KAN within the simplest point-cloud-based neural network, PointNet, we hope that the promising results of PointNet-KAN motivate future efforts to embed KANs into more advanced architectures.

While ModelNet40 (Wu et al., 2015) is a widely recognized benchmark for evaluating and comparing different methods for classification tasks, this dataset only contains synthetic data. To further assess the robustness and real-world applicability of PointNet-KAN, we extended our evaluation to the ScanObjectNN (Uy et al., 2019) dataset, which comprises real-world data. The dataset includes approximately 15,000 objects across 15 categories. Specifically, we utilized the PB_T50_RS variant of ScanObjectNN (Uy et al., 2019). The results are summarized in Table 2. Accordingly, PointNet-KAN (with $\alpha = \beta = 1$, $n = 4$) with normal vectors as input outperforms PointNet (Qi et al., 2017a), whereas without normal vectors, this performance advantage is not observed. We observed a similar trend in the classification task on ModelNet40 (Wu et al., 2015), as seen in Table 1. Incorporating normal vectors generally enhances performance by providing additional geometric information, as reported in prior studies (Qi et al., 2017b; Wang et al., 2019). However, it increases the computational cost of preprocessing. Furthermore, the method used to compute normal vectors might influence the performance.

Table 2: Classification results on ScanObjectNN, the PB_T50_RS dataset (Uy et al., 2019). In PointNet-KAN, the Jacobi polynomial degree is set to 4 (i.e., $n = 4$) with $\alpha = \beta = 1.0$.

| | Overall accuracy | Mean accuracy | bag | bin | box | cabinet | chair | desk | display | door | shelf | table | bed | pillow | sink | sofa | toilet |
|---|---|---|---|---|---|---|---|---|---|---|---|---|---|---|---|---|---|
| PointNet++ (Qi et al., 2017b) | 77.9 | 75.4 | 49.4 | 84.4 | 31.6 | 77.4 | 91.3 | 74 | 79.4 | 85.2 | 72.6 | 72.6 | 75.5 | 81 | 80.8 | 90.5 | 85.9 |
| DGCNN (Wang et al., 2019) | 78.1 | 73.6 | 49.4 | 82.4 | 33.1 | 83.9 | 91.8 | 63.3 | 77 | 89 | 79.3 | 77.4 | 64.5 | 77.1 | 75 | 91.4 | 69.4 |
| PointMLP (Ma et al., 2022) | 85.4 | 83.9 | - | - | - | - | - | - | - | - | - | - | - | - | - | - | - |
| PointNet (Qi et al., 2017a) | 68.2 | 63.4 | 36.1 | 69.8 | 10.5 | 62.6 | 89.0 | 50.0 | 73.0 | 93.8 | 72.6 | 67.8 | 61.8 | 67.6 | 64.2 | 76.7 | 55.3 |
| PointNet-KAN | 66.5 | 61.1 | 33.2 | 66.5 | 9.2 | 62.7 | 86.1 | 45.3 | 70.1 | 90.4 | 70.4 | 67.2 | 62.1 | 62.9 | 63.0 | 74.9 | 52.2 |
| PointNet-KAN with normal | 69.2 | 63.9 | 36.3 | 68.5 | 10.8 | 63.4 | 89.5 | 50.2 | 73.1 | 94.7 | 73.4 | 68.2 | 63.3 | 68.5 | 65.1 | 77.4 | 57.2 |

## 5.2 3D OBJECT PART SEGMENTATION

For the part segmentation task, we assess PointNet-KAN on the ShapeNet part dataset (Yi et al., 2016), which includes 16,881 shapes across 16 categories, with annotations for 50 distinct parts. The number of parts per category ranges from 2 to 6. We adhere to the official train, validation, and test splits as outlined in the literature (Chang et al., 2015; Qi et al., 2017a; Wang et al., 2019). In our experiment, we uniformly sample 2,048 points from each shape within a unit ball. The input features for PointNet-KAN consist solely of spatial coordinates, and normal vectors are not utilized (i.e., $d = 3$). The evaluation metric used is Intersection-over-Union (IoU) on points, as described by Qi et al. (2017a). Training details are provided in A.1. Qualitative results for part segmentation are shown in Fig. 2. The performance of PointNet-KAN compared to PointNet Qi et al. (2017a) is presented in Table 3. Accordingly, PointNet-KAN demonstrates competitive results compared to PointNet, with a mean IoU of 83.3% versus 83.7%. As shown in Table 3, for categories such as motorbike, pistol, and table, PointNet-KAN provides more accurate predictions than PointNet Qi et al. (2017a). Based on our machine learning experiments, adding normal vectors as input features does not improve the performance of PointNet-KAN. A comparison between the segmentation branch of PointNet-KAN,

Table 3: Mean IoU results for part segmentation on ShapeNet part dataset (Yi et al., 2016). In PointNet-KAN, the Jacobi polynomial degree is set to 2 (i.e., $n = 2$) with $\alpha = \beta = -0.5$. Results of other models allocated, Wu et al. (2014), 3DCNN (Qi et al., 2017a), Yi et al. (2016), PointNet (Qi et al., 2017a), DGCNN (Wang et al., 2019), KPConv (Thomas et al., 2019), TAP (Wang et al., 2023). PN-KAN stands for PointNet-KAN in this table.

| | Mean IoU | aero | bag | cap | car | chair | ear phone | guitar | knife | lamp | laptop | motor | mug | pistol | rocket | skate board | table |
|---|---|---|---|---|---|---|---|---|---|---|---|---|---|---|---|---|---|
| # shapes | | 2690 | 76 | 55 | 898 | 3758 | 69 | 787 | 392 | 1547 | 451 | 202 | 184 | 283 | 66 | 152 | 5271 |
| Wu et al. | - | 63.2 | - | - | - | 73.5 | - | - | - | 74.4 | - | - | - | - | - | 51.2 | 74.8 |
| 3DCNN | 79.4 | 75.1 | 72.8 | 73.3 | 70.0 | 87.2 | 63.5 | 88.4 | 79.6 | 74.4 | 93.9 | 58.7 | 91.8 | 76.4 | 51.2 | 65.3 | 77.1 |
| Yi et al. | 81.4 | 81.0 | 78.4 | 77.7 | 75.7 | 87.6 | 61.9 | 92.0 | 85.4 | 82.5 | 95.7 | 70.6 | 91.9 | 85.9 | 53.1 | 69.8 | 75.3 |
| DGCNN | 85.2 | 84.0 | 83.4 | 86.7 | 77.8 | 90.6 | 74.7 | 91.2 | 87.5 | 82.8 | 95.7 | 66.3 | 94.9 | 81.1 | 63.5 | 74.5 | 82.6 |
| KPConv | 86.4 | 84.6 | 86.3 | 87.2 | 81.1 | 91.1 | 77.8 | 92.6 | 88.4 | 82.7 | 96.2 | 78.1 | 95.8 | 85.4 | 69.0 | 82.0 | 83.6 |
| TAP | 86.9 | 84.8 | 86.1 | 89.5 | 82.5 | 92.1 | 75.9 | 92.3 | 88.7 | 85.6 | 96.5 | 79.8 | 96.0 | 85.9 | 66.2 | 78.1 | 83.2 |
| PointNet | 83.7 | 83.4 | 78.7 | 82.5 | 74.9 | 89.6 | 73.0 | 91.5 | 85.9 | 80.8 | 95.3 | 65.2 | 93.0 | 81.2 | 57.9 | 72.8 | 80.6 |
| PN-KAN | 83.3 | 81.0 | 76.8 | 79.8 | 74.6 | 88.7 | 65.4 | 90.9 | 85.3 | 79.9 | 95.0 | 65.3 | 93.0 | 83.0 | 54.3 | 71.9 | 81.6 |

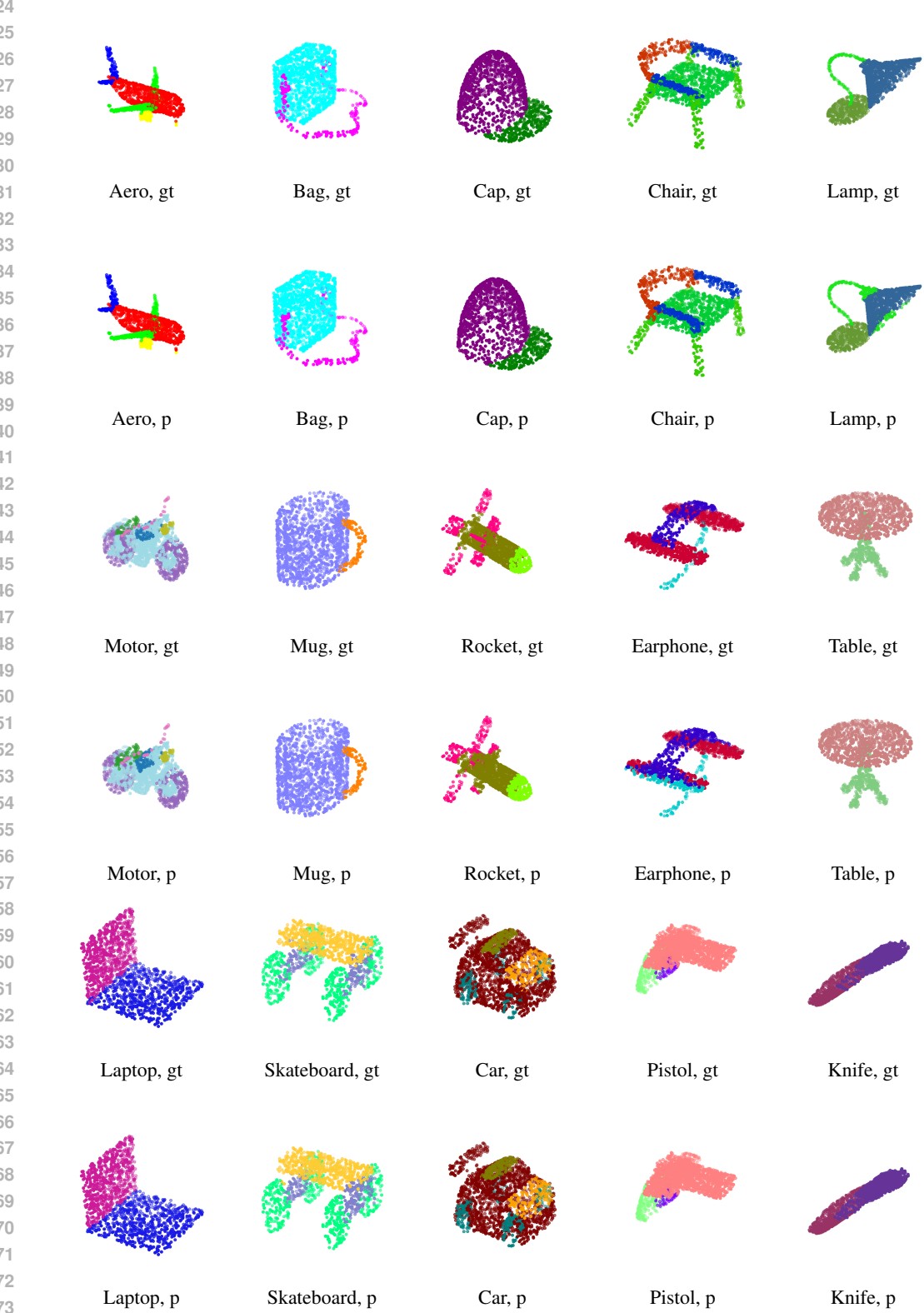

Figure 2: A few qualitative results obtained by PointNet-KAN for part segmentation on the ShapeNet Part dataset (Yi et al., 2016). The results correspond to PointNet-KAN using a Jacobi polynomial of degree 2 with $\alpha = \beta = -0.5$. In the labels, 'gt' represents the ground truth, and 'p' represents prediction.

Table 4: Effect of Jacobi polynomial degree on classification performance of PointNet-KAN with the choice of $\alpha = \beta = 1.0$ on ModelNet40 (Wu et al., 2015). Normal vectors are included as part of the input features.

| Jacobi polynomial degree ($n$) | 2 | 3 | 4 | 5 | 6 |
|---|---|---|---|---|---|
| Number of trainable parameters | 620928 | 823680 | 1026432 | 1229184 | 1431936 |
| Mean class accuracy | 86.7 | 87.0 | 87.2 | 86.8 | 86.1 |
| Overall accuracy | 89.9 | 90.4 | 90.5 | 89.9 | 89.9 |

Table 5: Effect of the choice of $\alpha$ and $\beta$ in Jacobi polynomials on the classification performance of PointNet-KAN, using a polynomial of degree 2 (i.e., $n = 2$ in Eq. 3), on ModelNet40 (Wu et al., 2015). Note that $\alpha = \beta = 0$ corresponds to the Legendre polynomial, $\alpha = \beta = -0.5$ corresponds to the Chebyshev polynomial of the first kind, $\alpha = \beta = 0.5$ corresponds to the Chebyshev polynomial of the second kind, and, in general, $\alpha = \beta$ corresponds to the Gegenbauer polynomial. Normal vectors are included as part of the input features.

| Polynomial type | $\alpha = \beta = 0$ | $\alpha = \beta = -0.5$ | $\alpha = \beta = 0.5$ | $\alpha = \beta = 1$ | $2\alpha = \beta = 2$ | $\alpha = 2\beta = 2$ |
|---|---|---|---|---|---|---|
| Mean class accuracy | 85.6 | 86.0 | 86.7 | 86.7 | 85.4 | 86.2 |
| Overall accuracy | 89.5 | 89.9 | 90.1 | 89.9 | 89.4 | 89.8 |

shown in Fig. 1, and the part segmentation branch of PointNet, shown in Fig. 9 of Qi et al. (2017a), highlights the simplicity of the PointNet-KAN architecture, which consists of only 4 layers and uses a single local feature, whereas PointNet has 11 layers and uses 5 local features. Additionally, while PointNet includes input and feature transform networks, the PointNet-KAN architecture does not. Overall, PointNet-KAN outperforms earlier methodologies such as those in Wu et al. (2014), 3DCNN (Qi et al., 2017a), and Yi et al. (2016). However, more recent architectures, including DGCNN (Wang et al., 2019), KPConv (Thomas et al., 2019), and TAP (Wang et al., 2023), surpass PointNet-KAN. As discussed in Sect. 5.1, incorporating KANs into the core of these networks as a replacement for MLPs could potentially enhance their performance.

## 5.3 ABLATION STUDIES

**Influence of polynomial type and polynomial degree**  Concerning the classification task discussed in Sect. 5.1, Table 4 illustrates the effect of varying the polynomial degree from 2 to 6, with $\alpha = \beta = 1$ held constant. While increasing the degree does not significantly affect accuracy, it does increase the number of trainable parameters. Moreover, Table 5 reports the results of varying $\alpha$ and $\beta$ with a fixed polynomial degree of 2, showing that different Jacobi polynomial types do not significantly impact performance. Concerning the segmentation task discussed in Sect. 5.2, we investigate the effect of the Jacobi polynomial degree and the roles of $\alpha$ and $\beta$ on performance. The results are tabulated in Table 6 and 7. Similar to the classification task discussed in Sect. 5.1, no significant differences are observed. As shown in Table 6, increasing the degree of the Jacobi polynomial does not improve prediction accuracy. According to Table 7, the best performance is achieved with the Chebyshev polynomial of the first kind when $\alpha = \beta = -0.5$.

**Influence of the size of tensors and global feature**  We investigate the effect of the size of the tensor **A** (see Eq. 2) and, consequently, the size of the global feature on prediction accuracy. In the classification branch (see Fig. 1), choosing the shared KAN layer with the size of 1024 (i.e., $\mathbf{A}_{1024 \times 6}$ and global feature size of 1024) and 2048 (i.e., $\mathbf{A}_{2048 \times 6}$ and global feature size of 2048) results in the overall accuracy of 89.7% and 90.3%, respectively, for the ModelNet40 (Wu et al., 2015) benchmark. In the segmentation branch (see Fig. 1), there are four shared KAN layers, each corresponding to a tensor. From left to right, we refer to them as **B**, **C**, **D**, and **E**. For example, selecting the sets $\mathbf{B}_{128 \times 3}$, $\mathbf{C}_{1024 \times 128}$, $\mathbf{D}_{128 \times 1153}$, $\mathbf{E}_{50 \times 128}$ and $\mathbf{B}_{384 \times 3}$, $\mathbf{C}_{3072 \times 384}$, $\mathbf{D}_{3457 \times 3457}$, $\mathbf{E}_{50 \times 384}$, respectively, results in a mean IoU of 82.6% and 82.2% for the ShapeNet part (Yi et al., 2016) benchmark. Note that the size of the global feature in the segmentation branch is determined by the number of rows ($d_{\text{output}}$) in tensor **C**.

Table 6: Mean IoU results of PointNet-KAN for part segmentation on ShapeNet part dataset (Yi et al., 2016) for different Jacobi polynomial degrees ($n$) with $\alpha = \beta = 1$.

| | Mean IoU | aero | bag | cap | car | chair | ear phone | guitar | knife | lamp | laptop | motor | mug | pistol | rocket | skate board | table |
|---|---|---|---|---|---|---|---|---|---|---|---|---|---|---|---|---|---|
| # shapes | | 2690 | 76 | 55 | 898 | 3758 | 69 | 787 | 392 | 1547 | 451 | 202 | 184 | 283 | 66 | 152 | 5271 |
| $n = 2$ | 82.8 | 81.1 | 76.8 | 78.7 | 74.4 | 88.4 | 64.8 | 90.5 | 84.5 | 78.8 | 95.0 | 66.9 | 93.0 | 82.3 | 56.8 | 73.5 | 80.7 |
| $n = 3$ | 81.8 | 80.0 | 76.3 | 79.6 | 72.1 | 88.0 | 69.4 | 89.0 | 83.0 | 79.4 | 95.0 | 61.5 | 91.3 | 81.0 | 55.3 | 70.0 | 79.0 |
| $n = 4$ | 82.4 | 81.2 | 71.2 | 75.6 | 70.7 | 87.9 | 68.3 | 90.0 | 81.8 | 78.4 | 94.0 | 60.7 | 90.7 | 80.1 | 51.3 | 70.8 | 81.7 |
| $n = 5$ | 80.7 | 78.2 | 72.0 | 79.0 | 67.8 | 87.5 | 68.9 | 87.6 | 81.3 | 76.6 | 94.5 | 60.8 | 88.0 | 81.0 | 47.3 | 69.3 | 79.3 |
| $n = 6$ | 82.2 | 80.5 | 70.8 | 78.0 | 71.7 | 87.5 | 62.5 | 88.0 | 82.7 | 76.8 | 94.6 | 62.8 | 92.0 | 78.9 | 48.7 | 65.9 | 81.6 |

**Robustness** Figure 3 shows the overall accuracy on the ModelNet40 (Wu et al., 2015) benchmark when input points from the test set are randomly dropped. PointNet-KAN (with $\alpha = \beta = 1$, $n = 4$) demonstrates relatively stable performance as the number of points decreases from 1024 to 128, with accuracy gradually dropping from 90.5% to 83.7% when using normal vectors ($d = 6$), and from 87.5% to 77.5% without normal vectors ($d = 3$). Interestingly, PointNet-KAN shows stronger stability compared to other models (Qi et al., 2017a;b; Wang et al., 2019), as indicated in Fig. 3. We further investigate the robustness of PointNet-KAN (with $\alpha = \beta = 1$, $n = 4$) compared to PointNet (Qi et al., 2017a) under input point perturbations using Gaussian noise, focusing on overall accuracy for the ModelNet40 (Wu et al., 2015) test case, as illustrated in Fig. 4. Similar levels of robustness are observed between PointNet-KAN and PointNet (Qi et al., 2017a), with PointNet-KAN showing slightly greater resilience. Comparing PointNet-KAN with and without normal

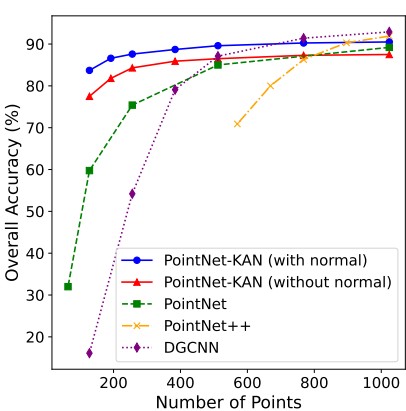

Figure 3: Robustness test for PointNet-KAN on the ModelNet40 (Wu et al., 2015) test case, where input points are randomly dropped. See text for details.

vectors, we observe that when normal vectors are included, PointNet-KAN demonstrates greater robustness as the standard deviation of the Gaussian noise increases up to 0.06. However, beyond a standard deviation of 0.06, both methods exhibit roughly the same performance, indicating that the inclusion of normal vectors no longer provides a significant advantage.

**Influence of input and feature transform networks and deeper architectures** In Sect. 5.1 and Sect. 5.2, we pointed out that PointNet-KAN is effective, despite its simple and shallow architecture, and the absence of input and feature transform networks. A question arises: if such a simple structure performs well, why not improve PointNet-KAN's performance by deepening the network and adding input and feature transform networks to achieve even better results? To answer this question, a straightforward approach is to replace all MLPs in the PointNet architecture (see Fig. 2 of Qi et al. (2017a) for the classification branch and Fig. 9 of Qi et al. (2017a) for the segmentation branch) with KAN to create an equivalent model. We conduct this experiment as follows. We utilize KAN layers with a Jacobi polynomial degree of 2 (i.e., $n = 2$) and parameters $\alpha = \beta = 1$. The size of the sequential KAN layers is chosen to match the correspond-

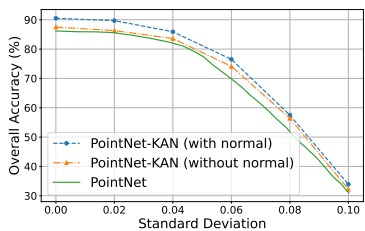

Figure 4: Robustness test for PointNet-KAN on the ModelNet40 (Wu et al., 2015) test case, where Gaussian noise with varying standard deviations is independently added to each point to perturb it. See text for details.

ing size of the MLPs in PointNet, such as (64, 64), (64, 128, 1024), and so on, as illustrated in Qi et al. (2017a). To conserve space, we omit sketching the full network architecture again. Interestingly, the network's performance does not improve. The overall accuracy of classification on ModelNet40 (Wu et al., 2015) is 88.9% and the mean IoU on the ShapeNet part dataset (Yi et al., 2016) is 82.1%.

Table 7: Mean IoU results of PointNet-KAN for part segmentation on ShapeNet part dataset (Yi et al., 2016) for different values of $\alpha$ and $\beta$. In PointNet-KAN, the Jacobi polynomial degree is set to 2 (i.e., $n = 2$). Note that $\alpha = \beta = 0$ corresponds to the Legendre polynomial, $\alpha = \beta = -0.5$ corresponds to the Chebyshev polynomial of the first kind, $\alpha = \beta = 0.5$ corresponds to the Chebyshev polynomial of the second kind, and, in general, $\alpha = \beta$ corresponds to the Gegenbauer polynomial.

| | Mean IoU | aero | bag | cap | car | chair | ear phone | guitar | knife | lamp | laptop | motor | mug | pistol | rocket | skate board | table |
|---|---|---|---|---|---|---|---|---|---|---|---|---|---|---|---|---|---|
| # shapes | | 2690 | 76 | 55 | 898 | 3758 | 69 | 787 | 392 | 1547 | 451 | 202 | 184 | 283 | 66 | 152 | 5271 |
| $\alpha = \beta = 0$ | 83.1 | 82.0 | 73.5 | 80.2 | 75.4 | 88.5 | 68.9 | 90.4 | 83.9 | 80.6 | 95.2 | 65.3 | 92.7 | 81.2 | 56.9 | 72.4 | 80.9 |
| $\alpha = \beta = -0.5$ | 83.3 | 81.0 | 76.8 | 79.8 | 74.6 | 88.7 | 65.4 | 90.9 | 85.3 | 79.9 | 95.0 | 65.3 | 93.0 | 83.0 | 54.3 | 71.9 | 81.6 |
| $\alpha = \beta = 0.5$ | 81.7 | 80.5 | 74.9 | 78.9 | 69.3 | 87.5 | 66.3 | 89.5 | 84.1 | 77.3 | 95.0 | 64.5 | 92.0 | 81.7 | 53.1 | 71.3 | 79.7 |
| $\alpha = \beta = 1$ | 82.8 | 81.1 | 76.8 | 78.7 | 74.4 | 88.4 | 64.8 | 90.5 | 84.5 | 78.8 | 95.0 | 66.9 | 93.0 | 82.3 | 56.8 | 73.5 | 80.7 |
| $2\alpha = \beta = 2$ | 82.6 | 81.0 | 75.8 | 81.5 | 72.1 | 88.1 | 68.0 | 90.9 | 83.5 | 79.5 | 95.2 | 63.2 | 91.2 | 80.5 | 58.2 | 74.0 | 80.8 |
| $\alpha = 2\beta = 2$ | 82.5 | 81.0 | 73.3 | 82.4 | 71.6 | 88.3 | 68.5 | 90.7 | 84.3 | 79.3 | 95.4 | 64.2 | 91.3 | 81.9 | 54.6 | 70.4 | 80.5 |

# 6 RELATED WORK

Relevant work on KANs can be discussed from two perspectives. The first focuses on using KANs for classification and segmentation tasks in computer graphics and computer vision. For classification, researchers (Cheon, 2024; Bodner et al., 2024; Azam & Akhtar, 2024) have embedded KANs as a replacement for MLPs in various popular CNN-based neural networks for two-dimensional image classification, such as VGG16 (Simonyan & Zisserman, 2014), MobileNetV2 (Sandler et al., 2018), EfficientNet (Tan, 2019), ConvNeXt (Liu et al., 2022), ResNet-101 (He et al., 2016), and Vision Transformer (Dosovitskiy, 2020), and evaluated the performance of these networks with KANs. For 3D image segmentation tasks, KANs have been embedded into U-Net (Ronneberger et al., 2015) as a replacement for MLPs (Tang et al., 2024; Wu et al., 2024). However, no prior work has explored the use of KANs in point-cloud-based neural networks for 3D classification and segmentation of unordered point sets or evaluated their performance on complex benchmark datasets such as ModelNet40 (Wu et al., 2015) and the ShapeNet Part dataset (Yi et al., 2016). From the second perspective, KANs were originally constructed using B-spline as the basis polynomial (Liu et al., 2024), and researchers employed this type of polynomial for image classification and segmentation (Cheon, 2024; Bodner et al., 2024; Azam & Akhtar, 2024). However, studies have shown that B-splines are computationally expensive and pose difficulties in implementation (Howard et al., 2024; Rigas et al., 2024). To address these issues, recent advancements in scientific machine learning suggested the use of Jacobi polynomials as an alternative in KANs (SS, 2024; Seydi, 2024). Accordingly, Jacobi polynomials are not only easier to implement but also computationally more efficient. However, no prior work has explored the use of KANs with Jacobi polynomials in computer vision for classification and segmentation tasks.

# 7 SUMMARY

In this work, we proposed, for the first time, PointNet with shared KANs (i.e., PointNet-KAN) and compared its performance to PointNet with shared MLPs. Our results demonstrated that PointNet-KAN achieved competitive performance to PointNet in both classification and segmentation tasks, while using a simpler and much shallower network compared to the deep PointNet with shared MLPs. In our implementation of shared KAN, we compared various families of the Jacobi polynomials, including Lagrange, Chebyshev, and Gegenbauer polynomials, and observed no significant differences in performance among them. Additionally, we found that a polynomial degree of 2 was sufficient. We hope this work lays a foundation and offers insights for incorporating KANs, as an alternative to MLPs, into more advanced architectures for point cloud deep learning frameworks.

## REPRODUCIBILITY STATEMENT

The code is currently provided in a zip file as supplementary material and is accessible to the public. After the review process, we will make it available in a public repository as well.

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

## A    Supplementary materials

### A.1    Training Details

The models for both classification and part segmentation are implemented using PyTorch. For classification tasks, a batch size of 64 is used, while part segmentation uses a batch size of 32. The training process employs the Adam optimizer, configured with $\beta_1 = 0.9$, $\beta_2 = 0.999$, and $\hat{\epsilon} = 10^{-8}$. An initial learning rate of 0.0005 and 0.001 is chosen respectively for the classification and part segmentation tasks. To progressively decrease the learning rate during training, a learning rate scheduler is applied, which reduces the learning rate by a factor of 0.5 after every 20 epochs. The cross-entropy loss function is used. All experiments run on an NVIDIA A100 Tensor Core GPU with 80 GB of RAM.

### A.2    A More Advanced Network: Extending PointMLP with Shared KANs

We further explore the potential of shared KANs in point-cloud-based neural networks by integrating them into more advanced architectures. Among the advanced neural networks discussed earlier in this study is PointMLP (Ma et al., 2022). Here, we present PointKAN, a framework that reconstructs PointMLP (Ma et al., 2022) using shared KANs.

We briefly review the PointMLP (Ma et al., 2022) architecture, which is fundamentally built on Residual Point (ResP) blocks. These ResP blocks form the backbone of an extractor, denoted as $\Phi$. In the PointMLP (Ma et al., 2022) framework, each stage consists of two extractors and an aggregation function. The first extractor ($\Phi_{\text{pre}}$) learns features from the input and passes them to the aggregation function, which employs max pooling. The output of this function is then fed into the second extractor ($\Phi_{\text{pos}}$), which extracts aggregated features (see Eq. 4 in Ma et al. (2022)). As discussed by Ma et al. (2022), one may optionally increase the number of extractors in each stage. Multiple stages can be connected sequentially to increase the depth of PointMLP. After several sequential stages, the final output is connected to a classifier for predicting classification scores. To enhance efficiency and stability, PointMLP (Ma et al., 2022) uses a geometric affine module before passing input points to the first stage (see Fig. 1 and Fig. 6 in Ma et al. (2022)). For a more detailed explanation, we refer readers to Ma et al. (2022).

To construct PointKAN, we modify the ResP blocks in PointMLP (Ma et al., 2022) by incorporating two sequential shared KAN layers, with each layer followed by batch normalization. Similar to PointMLP (Ma et al., 2022), PointKAN uses four stages. Each stage contains two extractor components ($\Phi_{\text{pre}}$), followed by max pooling as the aggregation function, and then two extractor components ($\Phi_{\text{pos}}$). The same geometric affine module is employed, as there is no MLP component embedded in this module (see Eq. 5 in Ma et al. (2022)); hence, no modification is required. For the classifier, we use the one designed for PointNet-KAN, as depicted in Fig. 1. For a fair comparison between PointMLP and PointKAN, we use the same dimensionality for each layer as in PointMLP, as illustrated in Fig. 6 of Ma et al. (2022). For the KAN and shared KAN layers, we set the Jacobi polynomial degree to 4 ($n = 4$) with $\alpha = \beta = 1.0$.

Table 8: Classification results on ModelNet40 (Wu et al., 2015) and ScanObjectNN, the PB_T50_RS dataset (Uy et al., 2019). In PointKAN and PointNet-KAN, the Jacobi polynomial degree is set to 4 (i.e., $n = 4$) with $\alpha = \beta = 1.0$.

| Test case | ModelNet40 | ModelNet40 | ScanObjectNN | ScanObjectNN |
|---|---|---|---|---|
| | Mean class accuracy | Overall accuracy | Mean class accuracy | Overall accuracy |
| PointNet (Qi et al., 2017a) | 86.2 | 89.2 | 63.4 | 68.2 |
| PointNet-KAN | 87.2 | 90.5 | 63.9 | 69.2 |
| PointNet++ (Qi et al., 2017b) | - | 91.9 | 75.4 | 77.9 |
| DGCNN (Wang et al., 2019) | 90.7 | 93.5 | 73.6 | 78.1 |
| Point Transformer (Zhao et al., 2021) | 90.6 | 93.7 | - | - |
| ShapeLLM (Qi et al., 2024) | 94.8 | 95.0 | - | 95.2 |
| PointMLP (Ma et al., 2022) | 91.4 | 94.5 | 83.9 | 85.4 |
| PointKAN | 91.7 | 94.6 | 84.1 | 85.5 |

We conduct machine learning experiments on the classification task using ModelNet40 (Wu et al., 2015) and ScanObjectNN, the PB_T50_RS dataset (Uy et al., 2019). The results are tabulated in Table 8. Comparing PointKAN with PointNet-KAN, we observe a significant improvement in the prediction accuracy of PointKAN. This highlights the critical role of using a more advanced architecture in enhancing network performance. Improvements are evident for both ModelNet40 (Wu et al., 2015) and ScanObjectNN (Uy et al., 2019). When comparing PointKAN with PointMLP (Ma et al., 2022), the prediction accuracy of PointKAN exceeds that of PointMLP by 0.1% for overall accuracy on the ModelNet40 (Wu et al., 2015) test case. On the ScanObjectNN (Uy et al., 2019) benchmark, PointNet-KAN outperforms PointMLP (Ma et al., 2022), although their performances are highly competitive, as shown in Table 8. Based on these experiments, we conclude two findings. First, integrating shared KANs into both basic and advanced point-cloud deep learning frameworks leads to successful neural networks. Second, combining shared KANs with advanced neural networks has the potential to improve performance compared to their counterparts with shared MLPs.

