# OpenReview forum: "PointNet with KAN versus PointNet with MLP for 3D Classification and Segmentation of Point Sets"
_ICLR.cc/2025/Conference — Submitted to ICLR 2025_

### Official Review · Reviewer_bvRm · 2024-10-23

**Soundness:** 2
**Presentation:** 3
**Contribution:** 2
**Rating:** 5
**Confidence:** 4

**Summary:**

This paper is the first to introduce KAN in the 3D domain, replacing MLP with KAN based on PointNet. Experiments are conducted on both classification and segmentation tasks, providing guidance for the subsequent application of KAN in point cloud analysis tasks.

**Strengths:**

This paper is the first to apply KAN to point cloud analysis tasks. The article is logically coherent, with complete formulas, and includes a substantial amount of experimentation to demonstrate the effectiveness of the proposed method, providing valuable guidance for the future development of KAN in point clouds.

**Weaknesses:**

1. Some experimental results are concerning and lack detailed discussion or explanation. For instance, on ModelNet40, there is a significant performance difference for PointNet-KAN depending on whether the input includes normal vectors, suggesting that PointNet-KAN is highly sensitive to input variations. However, the authors did not provide a more thorough investigation or explanation regarding this sensitivity. Additionally, on ShapeNet Part, the performance of PointNet-KAN still falls short compared to PointNet. Furthermore, based on subsequent experimental results, it seems that merely increasing the scale of PointNet-KAN does not address this issue.

2. The robustness experiments only consider variations in point quantity and lack experimental results when increasing noise and rotation.

3. For the classification tasks, experiments were only conducted on the ModelNet40 dataset and the PointNet model. It is recommended to include experiments on the ScanObjectNN dataset as well. Additionally, since PointMLP is also based on MLP, it may be worthwhile to explore embedding KAN into PointMLP.

**Questions:**

1. It is suggested that the authors provide a more detailed discussion and analysis of the experimental results. For example, they should explore why the presence of normal vectors has such a significant impact on the performance of PointNet-KAN on ModelNet40 and why the performance on ShapeNet Part is inferior to that of PointNet.

2. It is recommend that the authors include experiments on the ScanObjectNN dataset to further demonstrate the applicability of PointNet-KAN. Additionally, conducting experiments that address the effects of rotation and noise would provide a more comprehensive assessment of the robustness of PointNet-KAN.

3. Is there more detailed data regarding the $\textbf{influence of the size of tensors and global features}$? This could provide insights into the scalability of PointNet-KAN to some extent.

4. For Figure 2, I suggest adding a visual comparison with other methods to demonstrate the effectiveness of PointNet-KAN.

---

> ### Author Response · Authors · 2024-11-25
> **[Responses to the Weaknesses (W3) and Questions (Q2-1) Identified by Reviewer bvRm][Part 1]**
>
> Below are our responses to the Weaknesses (W3) and Questions (Q2) raised by the reviewer bvRm.
>
> **W3** For the classification tasks, experiments were only conducted on the ModelNet40 dataset and the PointNet model. It is recommended to include experiments on the ScanObjectNN dataset as well. Additionally, since PointMLP is also based on MLP, it may be worthwhile to explore embedding KAN into PointMLP.
>
> **Q2** It is recommend that the authors include experiments on the ScanObjectNN dataset to further demonstrate the applicability of PointNet-KAN.
>
> We are very thankful to the reviewer for providing this comment. Following their advice, we have added an experiment using the ScanObjectNN dataset in the revised manuscript to further challenge PointNet-KAN on real-world datasets. The results are presented in Table 2, with the discussion and explanation provided on page 6. The new additions are written in blue for clarity.
>
> Concerning trying PointMLP with shared KANs, we highly appreciate this suggestion. As a general comment, since KANs are still new to the community, exploring their integration into other networks undoubtedly adds value and provides valuable insights. However, specifically about the scope of the current manuscript, the initial version of the manuscript states on page 2, lines 68 to 72:
>
> *Moreover, we focus on PointNet rather than more advanced versions to directly and explicitly investigate the effect of KANs on the network's performance. Using more complex versions of PointNet could introduce other factors that might obscure the direct influence of KANs, making it challenging to determine whether any performance changes are due to the KAN architecture or other components of the network.*

---

> > ### Author Response · Authors · 2024-11-25
> > **[Responses to the Weaknesses (W2) and Questions (Q2-2) Identified by Reviewer bvRm][Part 2]**
> >
> > Below are our responses to the Weaknesses (W2) and Questions (Q2-2) raised by the reviewer bvRm.
> >
> > **W2** The robustness experiments only consider variations in point quantity and lack experimental results when increasing noise and rotation.
> >
> > **Q2** Additionally, conducting experiments that address the effects of rotation and noise would provide a more comprehensive assessment of the robustness of PointNet-KAN.
> >
> > We appreciate these comments. Following the reviewer’s recommendation, we have added a study on the effect of adding Gaussian noise to the revised manuscript. Specifically, we included Figure 4 along with its explanation and discussion in Section 5.3 (Ablation Studies: Robustness). The new additions are highlighted in blue for clarity.
> >
> > Concerning the study of rotation, we agree with the reviewer that it is indeed a challenging task. However, the original PointNet (and even its advanced versions, such as PointNet++) is not rotation-invariant, as it lacks the mathematical framework to support this property. As demonstrated in other studies, the performance of PointNet/PointNet++ drops significantly when subjected to rotations during the testing process. For instance, please refer to Table 1 in the following manuscript:
> >
> > https://arxiv.org/pdf/2010.03318
> >
> > In line with this suggestion, we conducted an additional experiment (not included in the manuscript) to investigate the effect of rotation-based data augmentation. Specifically, we applied random rotations along the z-axis during training for the classification task. However, we did not observe any improvement in the accuracy of PointNet-KAN compared to training without this form of data augmentation.

---

> ### Author Response · Authors · 2024-11-26
> **[Responses to the Weaknesses (W1) and Questions (Q1) Identified by Reviewer bvRm][Part 3]**
>
> Below are our responses to the Weaknesses (W1) and Questions (Q1) raised by the reviewer bvRm.
>
> **W1** Some experimental results are concerning and lack detailed discussion or explanation. For instance, on ModelNet40, there is a significant performance difference for PointNet-KAN depending on whether the input includes normal vectors, suggesting that PointNet-KAN is highly sensitive to input variations. However, the authors did not provide a more thorough investigation or explanation regarding this sensitivity. Additionally, on ShapeNet Part, the performance of PointNet-KAN still falls short compared to PointNet. Furthermore, based on subsequent experimental results, it seems that merely increasing the scale of PointNet-KAN does not address this issue.
>
> **Q1** It is suggested that the authors provide a more detailed discussion and analysis of the experimental results. For example, they should explore why the presence of normal vectors has such a significant impact on the performance of PointNet-KAN on ModelNet40 and why the performance on ShapeNet Part is inferior to that of PointNet.
>
> We highly appreciate the comments, questions, and suggestions by the reviewer. Concerning the effect of adding normal vectors, generally speaking, it is expected that incorporating additional information such as color, normal vectors, or increasing the number of input points can improve *overall* accuracy by approximately 3% to 4%. For instance, as reported in the PointNet++ manuscript, the overall accuracy for the classification task on ModelNet40 increases from 90.7% to 91.9% when normal vectors are included in the input. Please see Table 2 of the PointNet++ article for the official report:
>
> https://arxiv.org/pdf/1706.02413
>
> Additionally, PointNet++ uses 2048 points as input, compared to 1024 points for PointNet, further illustrating the impact of the number of input points on performance.
>
> As another example from the literature, the performance of DGCNN improves from 92.9% to 93.5% when the number of input points is increased from 1024 to 2048. Please refer to Table 2 of the following manuscript for the official report:
>
> https://arxiv.org/pdf/1801.07829
>
> Similarly, for PointNet-KAN, we observe a 3% increase in overall accuracy when normal vectors are added as inputs for the classification task on ModelNet40. For the new experiment added on ScanObjectNN (in Table 2), we observe an increase from 66.5 to 69.2 by including the normal vector. In the revised version of the manuscript, we have included the following paragraph on Page 6 (the last paragraph of Sect 5.1):
>
> *Accordingly, PointNet-KAN (with α = β = 1, n = 4) with normal vectors as input outperforms PointNet (Qi et al., 2017a), whereas without normal vectors, this performance advantage is not observed. We observed a similar trend in the classification task on ModelNet40 (Wu et al., 2015), as seen in Table 1. Incorporating normal vectors generally enhances performance by providing additional geometric information, as reported in prior studies (Qi et al., 2017b; Wang et al., 2019). However, it increases the computational cost of preprocessing. Furthermore, the method used to compute normal vectors might influence the performance.*
>
> Another possible reason is the recently reported tendency of KANs to overfit compared to MLPs. We conjecture that this characteristic might also influence the results of PointNet-KAN. Without normal vectors, the performance of PointNet-KAN is lower than that of PointNet. However, when normal vectors are added as inputs, the input features become more distinguishable for PointNet-KAN. This reduces the likelihood of overfitting on the training data and ultimately leads to improved performance. For the discussion about the prone of PointNet-KAN for overfitting please see the following article, for instance,
>
> https://arxiv.org/pdf/2411.06078 (SOMVANSHI et al. 2024)
>
> Concerning the comparison between PointNet-KAN and PointNet for part segmentation on the ShapeNet Part dataset, we would first like to emphasize our commitment to reporting honest results. As observed, the mean IoU is 83.3 for PointNet-KAN and 83.7 for PointNet, meaning that PointNet performs 0.4% better than PointNet-KAN. We conjecture that this slight difference may be attributed to PointNet-KAN’s tendency to overfit, as discussed regarding the effect of normal vectors. However, when examining individual object categories (please see Table 3), PointNet-KAN performs better than PointNet for "table," "motor," and "pistol,". Moreover, both PonintNet-KAN and PointNet achieve equal performance for "mug." This indicates that PointNet-KAN does not always perform worse than PointNet for part segmentation tasks. Furthermore, the impact of PointNet-KAN’s hyperparameters, such as the degree of the polynomial and the value of \alpha and \beta (which determine the type of polynomial), have been explored in Table 5 and Table 6, respectively.

---

> > ### Author Response · Authors · 2024-11-26
> > **[Responses to the Weaknesses Questions (Q3 and Q4) Identified by Reviewer bvRm][Part 4]**
> >
> > Below are our responses to the Questions (Q3 and Q4) raised by the reviewer bvRm.
> >
> > **Q3** Is there more detailed data regarding the influence of the size of tensors and global features? This could provide insights into the scalability of PointNet-KAN to some extent.
> >
> > Thank you for asking these questions. Beyond what we included in the manuscript, we can share with the reviewer that increasing the size of the global feature beyond 3072 for the classification task and 5120 for the part segmentation task does not improve the accuracy of PointNet-KAN for the current test cases discussed in the manuscript.
> >
> > Regarding scalability, we found the observations discussed in Section 5.3 (Ablation Studies: Influence of Input and Feature Transform Networks and Deeper Architectures) to be more insightful. Accordingly, making PointNet-KAN deeper and adding T-Nets (input and feature transforms) does not lead to an increase in accuracy. For further comparison, please compare Figure 1 of our manuscript with Figure 2 and Figure 9 of the original PointNet article:
> >
> > https://arxiv.org/pdf/1612.00593
> >
> > **Q4** For Figure 2, I suggest adding a visual comparison with other methods to demonstrate the effectiveness of PointNet-KAN.
> >
> > We highly appreciate the reviewer’s suggestion. A visual comparison can be generally helpful. However, for the part segmentation tasks, determining whether a part is predicted correctly or incorrectly and comparing it with other methods can be challenging to discern with the naked eye when presented in the manuscript. Specially since we cannot provide very high quality photos to avoid making the file size of the manuscript very big. All in all, we have already provided a comparison with the ground truth in Figure 2 of the manuscript, for the part segmentation task.
> >
> > We again appreciate the time and effort of the reviewer, as well as all the suggestions and comments, which have led to an improvement in the quality of this manuscript.

---

> ### Comment · Reviewer_bvRm · 2024-11-27
>
> Thank you for your response. However, due to the lack of experiments on modern architectures, I still have concerns about whether KAN has further potential applications in 3D point cloud analysis. Therefore, I will maintain my score as it is.

---

> ### Author Response · Authors · 2024-11-28
> **A more advanced point-cloud-based neural network is added to the revised version.**
>
> Below are our responses to the Weaknesses (W3) raised by the reviewer bvRm.
>
> **W3** For the classification tasks, experiments were only conducted on the ModelNet40 dataset and the PointNet model. It is recommended to include experiments on the ScanObjectNN dataset as well. Additionally, since PointMLP is also based on MLP, it may be worthwhile to explore embedding KAN into PointMLP.
>
> We had already added the experiments on the ScanObjectNN dataset. Following your advice for testing more advanced point-cloud-based neural networks (such as PointMLP as you suggested), we examine embedding the KAN and shared KAN layers into **PointMLP**.
>
> https://arxiv.org/pdf/2202.07123 (Ma et al.)
>
> We added the results to the Appendix in the revised version of the manuscript.
>
> We hope that this experiment as well as our previous answers addressed your concerns.
>
> We appreciate your time and looking forward to reevaluating our work. Thank you.

---

> > ### Author Response · Authors · 2024-12-02
> > **[Request for Reevaluation of Submission: PointNet with KAN]**
> >
> > Dear Reviewer bvRm,
> >
> > As the discussion phase will end soon, we kindly request that you reevaluate our work, considering the responses we have provided to your questions and concerns, as well as the additional experiments conducted in line with your suggestions.
> >
> > We sincerely appreciate your time and we look forward to hearing your thoughts.
> >
> > Warm regards,
> >
> > The Authors

---

> > > ### Comment · Reviewer_bvRm · 2024-12-02
> > >
> > > Thank you for your response. Our opinion is consistent with that of the other reviewers, who also question whether KAN is suitable for the modern architecture of point cloud models. Therefore, we will maintain the score.

---

> > > > ### Author Response · Authors · 2024-12-02
> > > > **[PointKAN, equivalent to PointMLP, as a modern architecture]**
> > > >
> > > > Dear Reviewer bvRm,
> > > >
> > > > Thanks for your comment. Just for clarification, in the revised version of the manuscript, we added **PointKAN** (equivalent to **PointMLP**), as you requested in your initial comments. **PointMLP** was proposed in 2022. I was wondering if you are considering **PointMLP** as a modern architecture.
> > > >
> > > > Here is the link to the **PointMLP** paper, for your reference:
> > > >
> > > > https://arxiv.org/abs/2202.07123
> > > >
> > > > Thank you.

---

### Official Review · Reviewer_y4gC · 2024-10-30

**Soundness:** 1
**Presentation:** 2
**Contribution:** 1
**Rating:** 3
**Confidence:** 4

**Summary:**

he paper integrates KAN into PointNet to propose a PointNet-KAN. It replaces MLPs with KAN and conducts validations on 3D object classification and part segmentation tasks. Compared with PointNet, the PointNet-KAN performs better on classification but maintains similar segmentation results on the ShapeNet part dataset. Overall, this work validates the KAN on 3D point domains but lacks novelty and contributions.

**Strengths:**

Integrating KAN into the 3D domain is interesting and may benefit the field of study.

**Weaknesses:**

1. Limited novelty and contribution.  This paper looks more like a technique report rather than a research paper. It naively uses KAN to replace the MLPs in PointNet without any technique improvements on KAN. It would be more preferable if the paper could adapt KAN in terms of some properties of 3D points, like irregularity and unorderness, to fit the nature of 3D points.

2. Lack of motivation to introduce KAN into 3D point cloud domain. In the Introduction part, the paper mentions that KAN can learn activation functions by itself but does not illustrate other benefits of introducing KAN into the 3D domain. Therefore, it is unclear why to integrate KAN into PointNet.

3. Experiment results are not satisfactory and need more improvements. In terms of Tables 1&2, PointNet-KAN can only marginally surpass PointNet on ModelNet classification and is even lower than PointNet on part segmentation tasks.These weak results fail to validate the effectiveness of the proposed method.

**Questions:**

If the paper aims to validate the effectiveness of replacing KAN with MLPs, how about trying more network architectures?

**Details Of Ethics Concerns:**

null

---

> ### Author Response · Authors · 2024-11-25
> **[Responses to the Weaknesses (W1 and W2) Identified by Reviewer y4gC][Part 1]**
>
> Below is a list of our responses to the Weaknesses (W1 and W2) raised by the reviewer y4gC.
>
> **W1** Limited novelty and contribution. This paper looks more like a technique report rather than a research paper. It naively uses KAN to replace the MLPs in PointNet without any technique improvements on KAN. It would be more preferable if the paper could adapt KAN in terms of some properties of 3D points, like irregularity and unorderness, to fit the nature of 3D points.
>
> **W2** Lack of motivation to introduce KAN into 3D point cloud domain. In the Introduction part, the paper mentions that KAN can learn activation functions by itself but does not illustrate other benefits of introducing KAN into the 3D domain. Therefore, it is unclear why to integrate KAN into PointNet.
>
>
> **Clarifying the Motivation and Contributions of This Work**
>
> We are very thankful to the reviewer for the comments. We would like to take this opportunity to further clarify the main motivation and contributions of the current manuscript.
>
> Kolmogorov-Arnold Networks (KANs) were proposed in April 2024. The authors of the original paper (Liu et al., 2024) claimed that KAN is an alternative to MLPs. Since then, KANs have been integrated into various deep learning frameworks, such as convolutional neural networks, graph neural networks, transformers, etc., and their performance has been evaluated. However, the effectiveness of KANs within point-cloud-based neural networks has not yet been explored. To address this, we incorporate KANs into PointNet (i.e., PointNet-KAN) for the first time to evaluate their performance on 3D point cloud classification and segmentation tasks. our work during the last summer started with finding the answer of some initial research questions.
>
> The first question that came to our mind was: Is it possible to implement PointNet with shared KANs rather than shared MLPs? Even if we could successfully implement them, a key concern was whether PointNet-KAN (i.e., PointNet with shared KANs) would show any performance at all or fail completely. The next question we had was: how do the results of part segmentation and classification of PointNet-KAN compare to those of PointNet with shared MLPs?
>
> To clarify these points for potential audiences, we have improved the abstract by adding a few sentences at the beginning that better highlight the motivation and significance of this work. The new additions are marked in blue in the revised version of the manuscript.
>
> Here is the updated version of the manuscript.
>
> *Abstract*
>
> *Kolmogorov–Arnold Networks (KANs) have recently gained attention as an alternative to traditional Multilayer Perceptrons (MLPs) in deep learning frameworks. KANs have been integrated into various deep learning architectures such as convolutional neural networks, graph neural networks, and transformers, with their performance evaluated. However, their effectiveness within point-cloud-based neural networks remains unexplored. To address this gap, we incorporate KANs into PointNet for the first time to evaluate their performance on 3D point cloud classification and segmentation tasks. Specifically, we introduce PointNet-KAN, built upon two key components. First, it employs KANs instead of traditional MLPs. Second, it retains the core principle of PointNet by using shared KAN layers and applying symmetric functions for global feature extraction, ensuring permutation invariance with respect to the input features. In traditional MLPs, the goal is to train the weights and biases with fixed activation functions; however, in KANs, the goal is to train the activation functions themselves. We use Jacobi polynomials to construct the KAN layers. We extensively and systematically evaluate PointNet-KAN across various polynomial degrees and special types such as the Lagrange, Chebyshev, and Gegenbauer polynomials. Our results show that PointNet-KAN achieves competitive performance compared to PointNet with MLPs on benchmark datasets for 3D object classification and segmentation, despite employing a shallower and simpler network architecture. We hope this work serves as a foundation and provides guidance for integrating KANs, as an alternative to MLPs, into more advanced point cloud processing architectures.*

---

> ### Author Response · Authors · 2024-11-25
> **[Responses to the Weaknesses (W1 and W2) Identified by Reviewer y4gC][Part 2]**
>
> Below is a list of our responses to the remaining Weaknesses (W1 and W2) raised by the reviewer y4gC.
>
> **W1** Limited novelty and contribution. This paper looks more like a technique report rather than a research paper. It naively uses KAN to replace the MLPs in PointNet without any technique improvements on KAN. It would be more preferable if the paper could adapt KAN in terms of some properties of 3D points, like irregularity and unorderness, to fit the nature of 3D points.
>
> **W2** Lack of motivation to introduce KAN into 3D point cloud domain. In the Introduction part, the paper mentions that KAN can learn activation functions by itself but does not illustrate other benefits of introducing KAN into the 3D domain. Therefore, it is unclear why to integrate KAN into PointNet.
>
> **Clarifying the Technical Contributions and Modifications of KAN in This Study**
>
> As stated in the manuscript, the *original* PointNet is based on two fundamental theorems to ensure invariance with respect to input point permutations:
>
> 1.	It uses the concept of "shared" MLPs (please see Eq. 10 of the manuscript).
>
> 2.	It uses a symmetric function, such as the maximum function, to extract global features (please see Eq. 10 of the manuscript).
>
> It is important to clarify that the concept of **shared** MLPs is different from regular fully connected networks. The concept of "shared" MLPs is explained in detail in the following manuscript:
>
> Network in Network (Lin et al., 2014): https://arxiv.org/abs/1312.4400.
>
> For instance, one may see the implementation of shared MLPs in the following code for the original PointNet article
>
> https://github.com/charlesq34/pointnet/blob/master/models/pointnet_cls.py
>
> where they used tf_util.conv2d (in TensorFlow) to implement it.
>
> In this work, we introduce the concept of shared KANs for the first time and have successfully implemented the associated code, which is provided in the supplementary materials. This required modifications to the original version of KANs. As the reviewer correctly pointed out, shared KANs are essential for handling the permutation invariance of 3D point sets.
>
> Please also note that PointNet-KAN shown in Fig. 1 of the current manuscript is not merely a straightforward replacement of shared MLP with shared KAN. As discussed in Sections 1 and 4 of the manuscript, we demonstrate that PointNet with shared KAN requires a much shallower network. Furthermore, the number of neurons per layer, the size of the global feature, and other architectural details differ from the original PointNet. Additionally, in Section 5.3 (Ablation Studies), we show that incorporating T-Nets (i.e., input transforms and feature transforms) does not affect the accuracy of PointNet-KAN, which contrasts with the original PointNet.
>
> Please compare the architecture of PointNet-KAN shown in Figure 1 of our manuscript with the PointNet architecture depicted in Figures 2 and 9 of the original PointNet paper:
>
> https://arxiv.org/pdf/1612.00593
>
> By comparing these figures and their technical details, we hope that it becomes clear that PointNet-KAN is not merely a simple replacement of shared MLPs with shared KANs.
>
> To further address this question and concern raised by the reviewer, we refer to the initial version of the manuscript, lines 60 to 68 on page 2:
>
> *It is important to clarify that by embedding KANs into PointNet, we do not simply mean replacing every instance of MLPs with KANs. While such an approach could be considered a research case, our goal is to preserve and utilize the core principles upon which PointNet is built. First, we apply shared KANs, meaning that the same KANs are applied to all input points. Second, we utilize a symmetric function, such as the max function, to extract global features from the points. These two elements are fundamental to PointNet, and by maintaining them, we ensure that the network remains invariant to input permutations. Our objective is to propose a version of PointNet integrated with KANs that retains these two essential properties, which we refer to as PointNet-KAN throughout the rest of this article.*

---

> ### Author Response · Authors · 2024-11-25
> **[Responses to the Weaknesses (W3) Identified by Reviewer y4gC][Part 3]**
>
> Below are our responses to the Weaknesses (W3) raised by the reviewer y4gC.
>
> **W3** Experiment results are not satisfactory and need more improvements. In terms of Tables 1&2, PointNet-KAN can only marginally surpass PointNet on ModelNet classification and is even lower than PointNet on part segmentation tasks.These weak results fail to validate the effectiveness of the proposed method.
>
> We appreciate this comment and agree with the reviewer that PointNet-KAN without normal vectors as input performs worse than PointNet with shared MLPs. However, adding normal vectors as inputs improves its performance, surpassing the original PointNet for the classification task. This trend is also observed in the new test case on ScanObjectNN, which has been added to the revised manuscript.
>
> To add more experimental results, we have added an experiment using the ScanObjectNN dataset in the revised manuscript to further challenge PointNet-KAN on real-world datasets. The results are presented in Table 2, with the discussion and explanation provided on page 6. The new additions are written in blue for clarity.
>
> We reiterate that we have never claimed that PointNet-KAN outperforms PointNet with shared MLPs. As stated earlier, our goal is to conduct a meaningful comparison, particularly since it was initially nontrivial to determine whether PointNet with shared KAN would work at all. In this work, we aim to provide an honest assessment to ensure neither overrating nor underrating KAN’s capabilities.

---

> ### Author Response · Authors · 2024-11-28
> **[Responses to the Questions Identified by Reviewer y4gC][Revised][A more advanced point-cloud-based neural network is added to the revised version.]**
>
> Below are our responses to the Question (Q1) raised by the reviewer y4gC.
>
> **Q1** If the paper aims to validate the effectiveness of replacing KAN with MLPs, how about trying more network architectures?
>
> Thanks for the question and suggestion. Following your advice for testing more advanced point-cloud-based neural networks, we examine embedding the KAN and shared KAN layers into PointMLP. Here is the link to the paper:
>
> https://arxiv.org/pdf/2202.07123 (Ma et al. 2022)
>
> We added the results to the Appendix in the revised version of the manuscript.
>
> We hope that this experiment as well as our previous answers addressed your concerns.
>
> We appreciate your time and looking forward to reevaluating our work. Thank you.

---

> > ### Comment · Reviewer_y4gC · 2024-12-01
> >
> > Thank the authors for providing additional materials. However, the current version still lacks strong motivations and experimental results to support the integration of KAN into point-based methods. I'll keep my rating.

---

> > > ### Author Response · Authors · 2024-12-02
> > >
> > > Dear Reviewer y4gC,
> > >
> > > Thanks for your comment. In the revised version of the manuscript:
> > >
> > > 1) We added the benchmark of **ScanObjectNN**
> > >
> > > 2) We integrated *shared* KAN with a more advanced point-cloud neural network **PointMLP** (2022), and added the results to the manuscript in the Appendix.
> > >
> > > Here is the link to the **PointMLP** paper for your reference:
> > >
> > > https://arxiv.org/pdf/2202.07123 (Ma et al. 2022)
> > >
> > > We would be very thankful if you could please let us know why these new results are not still satisfactory and what your expectations are. Then, it might help us clarify things better.
> > >
> > > We highly appreciate your time.

---

### Official Review · Reviewer_A2LP · 2024-11-04

**Soundness:** 2
**Presentation:** 3
**Contribution:** 2
**Rating:** 3
**Confidence:** 3

**Summary:**

The paper replaces Multi-Layer Perceptrons (MLPs) in the PointNet model with Kolmogorov-Arnold Network layers (KANs) to create a point-cloud-based neural network for classification or segmentation tasks on unordered 3D point sets. The proposed approach uses Jacobi polynomials to construct PointNet-KAN and investigate its performance across different polynomial degrees. The paper also includes efforts to examine the effect of special cases of Jacobi polynomials, including Legendre polynomials, Chebyshev polynomials of the first and second kinds, and Gegenbauer polynomials. The paper supports the proposed method with extensive evaluations of PointNet-KAN hyperparameters, such as the degree and type of polynomial used in constructing KANs. PointNet-KAN is a shallower and simpler network compared to PointNet and achieves competitive performance.

**Strengths:**

+ The proposed approach accommodates Jocbi polynomials and special case polynomials and provides experimental results.
+ The paper is well-written and easy to understand.

**Weaknesses:**

- The proposed method evaluation is limited to synthetic datasets. Most recent state-of-the-art 3D Classification methods use real-world datasets such as ScanObjectNN.
- The time complexity of the pointNet-KAN model during training/testing is not discussed. It is unclear if it is worth replacing MLPs with KANs as the overall accuracy improvements are less significant (< 2%).
- The proposed approach is a straightforward replacement of MLPs by KANs. Although it has limited novelty for a paper in the ICLR main track, it would be a good contribution if submitted to one of the workshops.
- While qualitative results for part segmentation are provided for Jacobi polynomials, the results for special polynomials such as Legendre, Chebyshev, and Gegenbauer polynomials are not provided.
- Table 3 shows that the increase in Jacobi polynomial degree and the corresponding increase in the number of parameters seem to have marginal improvements in both mean and overall accuracy. So, it is unclear if the increase in accuracy due to the increase in Jacobi polynomial degree from 2 to 4 is due to the increase in the number of parameters only and not due to the proposed approach.
- The mean or overall accuracy for classification and segmentation tasks is lower than that of the original PointNet model (refer to Tables 1 and 2).

**Questions:**

Kindly provide experimental results to back the claim below.

"Using more complex versions of PointNet could introduce other factors that might obscure the direct influence of KANs, making it challenging to determine whether any performance changes are due to the KAN architecture or other network components."

---

> ### Author Response · Authors · 2024-11-25
> **A General Explanation to Reviewer A2LP**
>
> **Clarifying the Motivation and Contributions of This Study**
>
> We highly appreciate the comments provided by the reviewer. Before addressing the specific questions and concerns, we would like to take this opportunity to further clarify the main motivation and contributions of the current manuscript.
>
> The Kolmogorov-Arnold Network (KAN) was proposed in April 2024. The authors of the original paper (Liu et al., 2024) claimed that KAN is an alternative to MLPs. Since then, KANs have been integrated into various deep learning frameworks, such as convolutional neural networks, graph neural networks, transformers, etc., and their performance has been evaluated. However, the effectiveness of KANs within point-cloud-based neural networks has not yet been explored. To address this, we incorporate KANs into PointNet (i.e., PointNet-KAN) for the first time to evaluate their performance on 3D point cloud classification and segmentation tasks.
>
> We would like to emphasize that we do not claim PointNet-KAN outperforms PointNet with MLPs. Instead, our work during the last summer started with finding the answer of some initial research questions.
>
> The first question that came to our mind was: Is it possible to implement PointNet with shared KANs rather than shared MLPs? Even if we could successfully implement them, a key concern was whether PointNet-KAN (i.e., PointNet with shared KANs) would show any performance at all or fail completely.
>
> As stated in the manuscript, PointNet is based on two fundamental theorems to ensure invariance with respect to input point permutations:
>
> 1.	It uses the concept of "shared" MLPs (please see Eq. 10 of the manuscript).
>
> 2.	It uses a symmetric function, such as the maximum function, to extract global features (please see Eq. 10 of the manuscript).
>
> It is important to clarify that the concept of "shared" MLPs is different from regular fully connected networks. The concept of "shared" MLPs is explained in detail in the following manuscript:
> Network in Network (Lin et al., 2014): https://arxiv.org/abs/1312.4400.
>
> The next question we had was: how do the results of part segmentation and classification of PointNet-KAN compare to those of PointNet with shared MLPs?
>
> We believe that before our work, the answers to these questions were entirely unknown and nontrivial. Through this manuscript, we aimed to provide these answers. The title of the manuscript, “PointNet with KAN versus PointNet with MLP,” clearly reflects our primary goal: to propose a comparison between these two methods.
>
> We do not claim that PointNet-KAN outperforms PointNet with MLPs. Instead, we have presented honest results regarding the performance of both approaches. As shown in the tables of the manuscript—and as the reviewer has noted—there are indeed cases where PointNet with MLPs performs better than PointNet-KAN.
>
> To clarify these points for potential audiences, we have improved the abstract by adding a few sentences at the beginning that better highlight the motivation and significance of this work. The new additions are marked in blue in the revised version of the manuscript.
>
> Here is the updated version of the manuscript.
>
> *Abstract*
>
> *Kolmogorov–Arnold Networks (KANs) have recently gained attention as an alternative to traditional Multilayer Perceptrons (MLPs) in deep learning frameworks. KANs have been integrated into various deep learning architectures such as convolutional neural networks, graph neural networks, and transformers, with their performance evaluated. However, their effectiveness within point-cloud-based neural networks remains unexplored. To address this gap, we incorporate KANs into PointNet for the first time to evaluate their performance on 3D point cloud classification and segmentation tasks. Specifically, we introduce PointNet-KAN, built upon two key components. First, it employs KANs instead of traditional MLPs. Second, it retains the core principle of PointNet by using shared KAN layers and applying symmetric functions for global feature extraction, ensuring permutation invariance with respect to the input features. In traditional MLPs, the goal is to train the weights and biases with fixed activation functions; however, in KANs, the goal is to train the activation functions themselves. We use Jacobi polynomials to construct the KAN layers. We extensively and systematically evaluate PointNet-KAN across various polynomial degrees and special types such as the Lagrange, Chebyshev, and Gegenbauer polynomials. Our results show that PointNet-KAN achieves competitive performance compared to PointNet with MLPs on benchmark datasets for 3D object classification and segmentation, despite employing a shallower and simpler network architecture. We hope this work serves as a foundation and provides guidance for integrating KANs, as an alternative to MLPs, into more advanced point cloud processing architectures.*

---

> ### Author Response · Authors · 2024-11-25
> **[Responses to the Weaknesses Identified by Reviewer A2LP][Part 1]**
>
> Below is a list of our responses to the Weaknesses raised by the reviewer A2LP.
>
> **W1** The proposed method evaluation is limited to synthetic datasets. Most recent state-of-the-art 3D Classification methods use real-world datasets such as ScanObjectNN.
>
> We appreciate the reviewer’s comment. Following their advice, we have added an experiment using the ScanObjectNN dataset in the revised manuscript to further challenge PointNet-KAN on real-world datasets. The results are presented in Table 2, with the discussion and explanation provided on page 6. The new additions are written in blue for clarity.
>
> **W2** The time complexity of the pointNet-KAN model during training/testing is not discussed. It is unclear if it is worth replacing MLPs with KANs as the overall accuracy improvements are less significant (< 2%).
>
> Thank you for providing this comment. The time complexity of PointNet-KAN and PointNet is already presented in the last column of Table 1 (FLOPs per sample) in the initial submission of this manuscript. Furthermore, in the second paragraph of Section 5.1, near lines 262 to 264, the manuscript states:
>
> *From a time complexity perspective, the number of floating-point operations required for one forward pass of the PointNet-KAN model is significantly lower than that of PointNet, as shown in Table 1.*
>
> As an additional comment, based on the data in Table 1, we observe that PointNet-KAN is computationally less expensive than PointNet. However, we emphasize once again that the goal of this manuscript was to provide a comparison between these two methods to offer insights. The conclusion of this manuscript is not that PointNet-KAN is definitively better than PointNet with shared MLPs. We kindly request the reviewer to consider our intentions for writing this manuscript, as explained in the general comment titled “A General Explanation to Reviewer A2LP.” Thank you for your thoughtful consideration.
>
> **W3** The proposed approach is a straightforward replacement of MLPs by KANs. Although it has limited novelty for a paper in the ICLR main track, it would be a good contribution if submitted to one of the workshops.
>
> Thank you for providing this feedback and sharing your thoughts with us. For further clarification, please refer to our explanation at the top under the general comment titled “A General Explanation to Reviewer A2LP,” where we discuss the implementation of shared KAN and the rationale behind replacing MLPs with KANs. Please note that the concept of shared KAN is essential for PointNet to handle permutation invariance, as discussed in Section 3 of the manuscript.
>
> Please also note that the network shown in Fig. 1 of the manuscript, PointNet-KAN, is not merely a straightforward replacement of shared MLP with shared KAN. As discussed in Sections 1 and 4 of the manuscript, we demonstrate that PointNet with shared KAN requires a much shallower network. Furthermore, the number of neurons per layer, the size of the global feature, and other architectural details differ from the original PointNet. Additionally, in Section 5.3 (Ablation Studies), we show that incorporating T-Nets (i.e., input transforms and feature transforms) does not affect the accuracy of PointNet-KAN, which contrasts with the original PointNet.
>
> Please compare the architecture of PointNet-KAN shown in Figure 1 of our manuscript with the PointNet architecture depicted in Figures 2 and 9 of the original PointNet paper:
>
> https://arxiv.org/pdf/1612.00593
>
> By comparing these figures and its technical details, we hope that it becomes clear that PointNet-KAN is not merely a simple replacement of shared MLPs with shared KANs.
>
> To further address this question and concern raised by the reviewer, we refer to the initial version of the manuscript, lines 60 to 68 on page 2:
>
> *It is important to clarify that by embedding KANs into PointNet, we do not simply mean replacing every instance of MLPs with KANs. While such an approach could be considered a research case, our goal is to preserve and utilize the core principles upon which PointNet is built. First, we apply shared KANs, meaning that the same KANs are applied to all input points. Second, we utilize a symmetric function, such as the max function, to extract global features from the points. These two elements are fundamental to PointNet, and by maintaining them, we ensure that the network remains invariant to input permutations. Our objective is to propose a version of PointNet integrated with KANs that retains these two essential properties, which we refer to as PointNet-KAN throughout the rest of this article.*

---

> ### Author Response · Authors · 2024-11-25
> **[Responses to the Weaknesses Identified by Reviewer A2LP][Part 2]**
>
> Below is a list of our responses to the remaining Weaknesses raised by the reviewer A2LP.
>
> **W3** While qualitative results for part segmentation are provided for Jacobi polynomials, the results for special polynomials such as Legendre, Chebyshev, and Gegenbauer polynomials are not provided.
>
> Thank you for raising this concern. Please note that Jacobi polynomials form a family of polynomials, with specific types such as Legendre, Chebyshev, and Gegenbauer polynomials being special cases (or subsets) of Jacobi polynomials. By adjusting the parameters α\alphaα and β\betaβ, one can switch between these variants. As stated in the last paragraph of Section 2 of the manuscript (Page 3, lines 147–152):
>
> *Finally, setting $\alpha = \beta = 0$ yields the **Legendre polynomial**, while the **Chebyshev polynomials** of the first and second kinds are obtained with $\alpha = \beta = -0.5$ and $\alpha = \beta = 0.5$, respectively. Additionally, the **Gegenbauer (or ultraspherical) polynomials** arise when $\alpha = \beta$.*
>
> More specifically, Table 5 presents the quantitative results on how the choice of $\alpha$ and $\beta$ in Jacobi polynomials (i.e., different types of polynomials) affects the classification performance of PointNet-KAN. Similarly, Table 7 provides the mean IoU results of PointNet-KAN for part segmentation on the ShapeNet Part dataset for different values of $\alpha$ and $\beta$ (i.e., different types of polynomials). Therefore, the results for part segmentation using **Legendre, Chebyshev, and Gegenbauer polynomials** are included in the manuscript.
>
> **W4** Table 3 shows that the increase in Jacobi polynomial degree and the corresponding increase in the number of parameters seem to have marginal improvements in both mean and overall accuracy. So, it is unclear if the increase in accuracy due to the increase in Jacobi polynomial degree from 2 to 4 is due to the increase in the number of parameters only and not due to the proposed approach.
>
> Thanks for your comment. The goal of Table 3 (which is now Table 4, since we added a new table to the manuscript) was simply to provide the performance of PointNet-KAN for different degrees of Polynomial. As can be seen in Eq. 3 of the manuscript, the degree of the polynomial $n$ is a hyper-parameter of KAN and needs to be tune. Traditional MLPs do not have such as a hyperparameter. Hence, the degree of the polynomial as a hyperparameter is embedded in the proposed method. It has been discussed in detail in the first paragraph of Section 5.1: Ablation Studies, ''influence of polynomial type and polynomial degree.''
>
> **W5** The mean or overall accuracy for classification and segmentation tasks is lower than that of the original PointNet model (refer to Tables 1 and 2).
>
> We appreciate this comment and agree with the reviewer that PointNet-KAN without normal vectors as input performs worse than PointNet with shared MLPs. However, adding normal vectors as inputs improves its performance, surpassing the original PointNet. This trend is also observed in the new test case on ScanObjectNN, which has been added to the revised manuscript. Additionally, we have explained in the manuscript why incorporating normal vectors benefits PointNet-KAN, as we read in the manuscript (in the last paragraph of Sect 5.1 3D Object Classification)
>
> *Accordingly, PointNet-KAN (with $\alpha = \beta = 1$, $n = 4$) with normal vectors as input outperforms PointNet, whereas without normal vectors, this performance advantage is not observed. We observed a similar trend in the classification task on ModelNet40, as seen in Table 1. Incorporating normal vectors generally enhances performance by providing additional geometric information, as reported in prior studies. However, it increases the computational cost of preprocessing. Furthermore, the method used to compute normal vectors might influence the performance.*
>
> As another point, please refer to our explanation at the top, as the general comment titled “A General Explanation to Reviewer A2LP.” We reiterate that we have never claimed that PointNet-KAN outperforms PointNet with shared MLPs. As stated earlier, our goal is to conduct a meaningful comparison, particularly since it was initially nontrivial to determine whether PointNet with shared KAN would work at all. In this work, we aim to provide an honest assessment to ensure neither overrating nor underrating KAN’s capabilities.

---

> ### Author Response · Authors · 2024-11-28
> **[Responses to the Questions Identified by Reviewer A2LP][Revised][A more advanced point-cloud-based neural network is added to the revised version.]**
>
> Below are our responses to the Question raised by reviewer A2LP.
>
> **Q1** Kindly provide experimental results to back the claim below.
>
> "Using more complex versions of PointNet could introduce other factors that might obscure the direct influence of KANs, making it challenging to determine whether any performance changes are due to the KAN architecture or other network components."
>
> Thank you for addressing this question. To us, this sentence does not appear to be a claim. Rather, it simply conveys that we conducted our comparison with the original and basic PointNet to investigate the performance of PointNet-KAN. However, after receiving your comments, we added new experiments.
>
> Following your advice for testing more advanced point-cloud-based neural networks, we examine embedding the KAN and shared KAN layers into **PointMLP**. Here is the link to the paper:
>
> https://arxiv.org/pdf/2202.07123 (Ma et al. 2022)
>
> We added the results to the Appendix in the revised version of the manuscript.
>
> We hope that this experiment as well as our previous answers addressed your concerns.
>
> We appreciate your time and looking forward to reevaluating our work. Thank you.

---

> > ### Author Response · Authors · 2024-12-02
> > **[Request for Reevaluation of Submission][PointNet with KAN]**
> >
> > Dear Reviewer A2LP,
> >
> > With the discussion phase coming to a close, we kindly ask you to reevaluate our work, taking into account the responses we have provided to address your questions and concerns, as well as the additional experiments conducted based on your suggestions.
> >
> > We deeply appreciate your time and we look forward to your feedback.
> >
> > Best regards,
> >
> > The Authors

---

> > > ### Author Response · Authors · 2024-12-02
> > >
> > > Dear Reviewer A2LP,
> > >
> > > We would be very thankful if you could please reevaluate our work. Specifically, in the revised version of the manuscript:
> > >
> > > 1) We added the benchmark of **ScanObjectNN**
> > >
> > > 2) We integrated *shared* KAN with a more advanced point-cloud neural network **PointMLP** (2022), and added the results to the manuscript in the Appendix.
> > >
> > > Here is the link to the PointMLP paper for your reference:
> > >
> > > https://arxiv.org/pdf/2202.07123 (Ma et al. 2022)
> > >
> > > Much appreciate your time for considering our responses and the revised version.

---

### Author Response · Authors · 2024-11-26
**[Acknowledgment and Request for Reevaluation]**

We sincerely appreciate all the comments, questions, and concerns provided by the reviewers. We have thoroughly answered all the questions, addressed the concerns, and submitted a revised version of the manuscript. We would be grateful if the reviewers could kindly reevaluate our work accordingly. Thank you.

---

> ### Author Response · Authors · 2024-11-28
> **[2nd revision is submitted][Acknowledgment and Request for Reevaluation]**
>
> Dear Reviewers,
> We submitted the second revision of the manuscript to answer questions and address the concerns. We would be thankful if you could kindly reevaluate our submission.
> Thank you.

---

### Author Response · Authors · 2024-12-04
**[Kind Request to the Area Chair: Fair Assessment of Our Submission]**

Dear Area Chair,

We greatly appreciate the efforts of the reviewers in providing feedback on our submission and their valuable time. However, we have concerns that our submission did not receive adequate engagement during the discussion period, and critical aspects of our rebuttal appear to have been overlooked.

Below, we summarize the major additions made to our manuscript in response to the reviewers' requests:

1. We added experiments for classification on **ScanObjectNN**.

2. We integrated KAN into **PointMLP** in addition to its integration into **PointNet**.

Here is a summary of the reviewers' actions during the discussion period:

--> Reviewer **A2LP**: This reviewer never responded to our rebuttal or engaged in the discussion period.

--> Reviewer **y4gC**: Provided a single general comment:

*Thank the authors for providing additional materials. However, the current version still lacks strong motivations and experimental results to support the integration of KAN into point-based methods. I'll keep my rating.*

This feedback does not clarify what specific experiments or additional motivations were expected. Without further details, it was challenging to address these concerns effectively.

--> Reviewer **bvRm**: Requested results for **ScanObjectNN** and integration of KAN into a modern architecture such as **PointMLP**. We provided these results, yet the final comment stated:

*Thank you for your response. Our opinion is consistent with that of the other reviewers, who also question whether KAN is suitable for the modern architecture of point cloud models. Therefore, we will maintain the score.*

This comment appears to disregard our response and newly added experiments with **PointMLP**, which was previously mentioned as an example of a modern architecture by the reviewer.

We are concerned that these issues may have prevented a fair evaluation of our submission. Specifically, the reviewers’ lack of detailed feedback and engagement during the discussion period has left us unclear about their expectations or concerns.

We sincerely hope you can consider these points in your assessment to ensure a fair review of our work.

We highly appreciate your time and considering our concerns.

Best regards,

The Authors

---

### Meta-Review · Area_Chair_a9Hi · 2024-12-20

**Metareview:**

All reviewers unanimously provided negative feedback, primarily concerning why KAN can enhance point cloud parsing, both in terms of innovation and experimental aspects. Although the authors supplemented the relevant experiments as requested during the rebuttal process, the reviewers still held negative views. Their comments are presented as follows.

Reviewer A2LP: The evaluation is limited to synthetic datasets, with no experiments on real-world datasets like ScanObjectNN. The proposed approach lacks significant novelty, as it is primarily a straightforward replacement of MLPs with KANs. Additionally, the time complexity and computational efficiency of PointNet-KAN are not sufficiently explored.

Reviewer y4gC: The paper lacks methodological novelty and appears more like a technical report. The results fail to validate the effectiveness of KAN integration, as PointNet-KAN only marginally outperforms PointNet for classification and underperforms for segmentation. The motivation for introducing KAN into 3D domains is unclear, and the experimental results are not compelling.

Reviewer bvRm:  The experimental results are inconsistent, with PointNet-KAN performing worse than PointNet in segmentation and being highly sensitive to input variations like normal vectors. Robustness experiments are limited and do not address important factors like noise and rotation. The lack of experiments on modern architectures like PointMLP raises concerns about generalizability.

Two reviewers recommended rejection (scores of 3), and one reviewer provided a marginally below acceptance threshold score (5). The reviewers collectively question the contribution and practical relevance of the proposed method. Therefore, my recommendation is to reject the paper.

**Additional Comments On Reviewer Discussion:**

Although the authors supplemented the relevant experiments as requested during the rebuttal process, the reviewers still held negative views

---

### Decision · Program_Chairs · 2025-01-22

Reject